# A Cross-Sectional Study Exploring the Physical Activity Levels of Afghans and Other South Asian Youth in the UK

**DOI:** 10.3390/ijerph20021087

**Published:** 2023-01-07

**Authors:** Ayazullah Safi, Irfan Khawaja, Peter Collins, Tony Myers

**Affiliations:** 1Centre for Nutraceuticals, School of Life Sciences, University of Westminster, London W1W 6UW, UK; 2Department of Sport and Exercise, School of Health Sciences, Birmingham City University, Birmingham B15 3TN, UK; 3Faculty of Education Health and Wellbeing, University of Wolverhampton, Wolverhampton WV1 1LY, UK; 4Department of Social Science, Sport and Business, Newman University, Birmingham B32 3NT, UK

**Keywords:** physical activity level, British South Asian, young people, ethnic minority, health

## Abstract

Introduction: Participating in regular physical activity (PA) has numerous benefits, such as reducing obesity, chronic degenerative conditions, and depression. Despite many health-related benefits, physical inactivity is increasing in young people worldwide, especially in ethnic minority groups, such as British South Asians (BSAs). The aim of this study was to explore the PA levels of BSAs, specifically focusing on youth from Afghan, Pakistani, Bangladeshi, and Indian groups. Methods: A total of 191 (Afghans *N* = 44; Bangladeshi *N* = 39; Indian *N* = 56, Pakistani *N* = 52) youth from the West Midlands (UK) participated in this study (mean age 15.4 ± 0.5). The International Physical Activity Questionnaire—Short Form was used to measure PA levels. Data were modelled using a Bayesian approach to determine differences in PA levels. Results: The results indicated that 88.5% Afghans, 80% Bangladeshi, 78.6% Indians and 63% Pakistani reported engaging in <30 min of PA per day. Additionally, boys were more active than girls across all ethnic groups. Discussion: This study highlighted an alarmingly low proportion of young people from each BSA ethnic group meeting the PA guidelines. To the authors’ knowledge, this is the first study to explore and compare PA levels of the young British Afghan population, thus contributing novel findings to the area of BSA PA. Conclusion: Overall, the vast majority of BSA young people failed to meet the recommended PA guidelines of 60 min per day. Future research could utilise objective methods, such as Global Positioning System, pedometers and accelerometery to track and monitor PA levels, and could adopt an ecological approach to explore determinants of PA within each ethnic and gender group.

## 1. Introduction

Physical inactivity is a serious risk to health and has been identified as the fourth leading risk factor for mortality around the globe [1]. Participating in regular physical activity (PA) has physiological and psychological benefits, such as reducing the risk of heart disease, diabetes, obesity, depression, and anxiety [1,2,3]. Despite many health-related benefits associated with PA, physical inactivity is increasing in young people worldwide, especially in ethnic minority groups [4,5,6]. Moreover, evidence has outlined that there are a broad range of determinants impacting the likelihood of ethnic minority groups, such as South Asians, engaging in PA, especially socio-cultural and psychosocial barriers [7,8]. Evidence shows that British South Asian societies participate in lower levels of PA, consequently leading to serious health concerns within this population [7].

According to the latest national statistics in England, 68.5% of white English adults were reported as active, compared to just 51.9% of British South Asians (BSAs), who were the least active adult ethnic group. Moreover, the percentage of BSAs in England reported as inactive (34.5%) is higher than the national average (22.9%) or any other ethnic group [9]. When focusing on children and young people, research shows that only 44% in England are classified as active, and when focusing on young people aged 14–16 years-old specifically, only 41% are active [10]. When referring to different ethnicities, the 2020/21 statistics reveal that only 38.7% of BSA were active, which is considerably lower than their white British counterparts (47.7%) or the national average (44.6%) [9].

Moreover, research has suggested that BSAs in the United Kingdom (UK) are five times more likely to develop diabetes and have a two-fold higher risk of heart diseases compared to the general population [11]. While there seems to be some consensus about the lack of PA in the BSA groups as a whole, the evidence on differences in specific groups appears more equivocal. For example, previous research has revealed that the young Bangladeshi community engage in the least PA, followed by those of Pakistani and Indian descent [12]. In contrast, ref. [13] found no differences between Pakistani, Bangladeshi and Indian PA levels. Despite previous research exploring different BSA groups, including Bangladeshi, Indian and Pakistani communities [12,14], there appears to be limited studies investigating the UK Afghan population, which is a growing migrant population [15]. While Afghans share similarities to other BSA groups residing in the UK in terms of some cultural practices and religion, there are also likely to be differences in PA lifestyle [15].

Cultural expectations and acceptance of values of PA differs across cultures, countries, and continents. Individuals from countries where PA is regarded as important for health are more likely to engage in regular PA compared to cultures or countries where PA is not a social norm [15]. For example, cultural attitudes towards PA and environmental differences were postulated as explanations for British participants being more active compared to counterparts in Saudi Arabia [16]. Moreover, people from South Asian countries generally report lower levels of PA compared to European countries [17,18]. However, most of the existing literature considers BSAs as a homogeneous group [19,20]. Therefore, to understand the nuances of this global issue, it is vital to evaluate the PA levels across BSA young people heterogeneously. According to [21], some studies that explored ethnic differences in PA considered females only; thus, the interaction effect of BSA young people’s PA levels across gender remains unclear. The possibility of reporting separate results for gender, ethnicity and age groups could yield different results about this under-researched population [8,20].

South Asians are one of the largest and fastest-growing minority groups in the UK [22], and although the exact figure of Afghans in the UK is unknown, Afghans in particular are one of the most rapidly increasing populations in the UK, rising from 63,000 in 2011 to an estimated 79,000, with 49% being British nationals [23]. The Afghan population in the UK is expected to increase due to the collapse of its government in August 2021 and Taliban takeover. Since then, the UK has evacuated 17,000 individuals and is expected to resettle an additional 20,000 people over the next three years. Therefore, the current study is timely in aiming to explore the PA levels of the BSA community, specifically focusing on Afghans, Pakistani, Bangladeshi, and Indian groups. Additionally, this paper will explore differences according to gender.

## 2. Methods

### 2.1. Participants and Procedures

Institutional ethical approval was granted (Project ID number: 2015-12-08-1304075/2330) before secondary schools, religious places (such as Mosques and Gurdwaras), leisure centres, tuition academies, and community clubs in Birmingham (West Midlands) were contacted and invited to participate in the study. Participant and parental/legal guardian consent was received for 191 BSA young people aged 15 and 16 years of age to partake in the study. Participants provided details of their ethnic background and gender, which enabled analyses of these different groups to take place. Participant breakdown according to ethnic group and gender is provided in Table 1. According to census data, 6.8% of the population of England and Wales is South Asian, with 36.8% being Indian, 29.4% Pakistani, 11.8% Bangladeshi and 22% other (including Afghans) [24]. The study sample is broadly representative of these ethnic groups, with 29.3% Indians, 27.2% Pakistani, 20.4% Bangladeshi and 23.0% Afghans.

### 2.2. Measurement of PA

#### International Physical Activity Questionnaire—Short Form (IPAQ-SF)

The International Physical Activity Questionnaire—Short Form (IPAQ-SF) was used to measure the PA levels of young BSAs. The IPAQ is a standardised, valid, reliable, and culturally adaptable measurement tool that has been utilised widely across different cultural areas of the world [25,26,27].

Additionally, IPAQ is one of the most widely utilised PA questionnaires internationally [28,29,30,31,32]. Moreover, previous research has used and recognised IPAQ-SF as one of the most feasible, user-friendly, and cost-effective tools to assess PA levels [30,33,34,35]. The IPAQ-SF includes seven items and four subscales, which assess the frequency and duration of walking, moderate-intensity activities (e.g., activities including elevated heart rate and commencement of perspiration), vigorous intensity activities (e.g., activities dramatically increasing heart rate and breathing rate) and time spent sitting. Frequency (measured in days per week) and duration (time per day) were collected separately for each specific type of activity, with total hours engaged in for an activity being the product of the frequency and duration reported. The data collection period was one month during the summer term from secondary schools, religious places, leisure centres, and community clubs. The survey was distributed to participants during lessons and pre- and post-prayer times/activities, and administered by the lead researcher, who explained the purpose of the research and provided instructions on how to complete the survey. The lead researcher was present whilst participants completed the survey to offer additional support as required. 

Whilst the IPAQ-SF is widely utilised, there is a lack of a standardised approach to reporting the results, with a recent systematic review and meta-analysis by [35] recommending the assessment of moderate-to-vigorous PA (MVPA) due to its low bias risk and acceptable reliability and validity scores. Moreover, assessing minutes of MVPA enables the findings to be clearly and directly compared to the latest PA health guidelines, which indicate that children and young people should aim to be active for at least 60 min per day across the week [36]. However, focusing on MVPA in minutes alone does have its limitations, as it does not account for each person’s total energy expenditure since it lacks the sensitivity to account for the composition of MVPA (the proportion of moderate- and vigorous-intensity PA). Subsequently, for the purpose of rigour, clarity and transparency, the current study reports the total minutes per week of walking, moderate and vigorous-intensity activities, as well as the total minutes of MVPA per week and total metabolic equivalent of tasks (METs) in minutes per week. The classifications adopted for activity levels in this paper were based on 2 cut-off points of 30 min and 60 min per day of at least a moderate level of PA, as conducted in similar previous studies [16,37]. These cut-off points were applied to categorise young people’s PA in accordance with the UK’s current PA health guidelines [38].

### 2.3. Statistical Analysis

The descriptive statistics were calculated, including means, standard deviations, mean differences, and Cohen’s effect size, for vigorous PA, moderate PA, walking, combined MVPA and total daily METs of young BSA [39]. We modelled differences between young BSA groups using a Bayesian approach fitting models in R [40] with the Bayesian Regression Models using the Stan package [41]. Bayesian models are highly flexible, provide direct probabilities of differences, and help avoid some misunderstandings and assumptions of traditional *p*-values and confidence intervals [42,43]. Importantly for the present study, they do not require a randomised sample from the population of interest—something not attainable in cross-sectional studies. 

In each model of the differences in vigorous PA, moderate PA, walking and combined MVPA, the prior used for the intercept (*α*) was a student-t distribution with 3 degrees of freedom (ν), the location parameter (μ) the median of the response variable, and the scale parameter (σ) the median absolute deviation of the response variable. For *β*, the prior was a wide normal prior (μ, σ). BSA groups were fitted as categorical predictors, with the Afghan group used as the reference category (see Equation (1)).
*y*_i_∼Normal (μ_i_, σe)(1)
μ_i_ = *α* + *β*_1_(Ethnicity_Bangladeshi_) + *β*_2_(Ethnicity_Indian_) + *β*_3_(Ethnicity_Pakistani_)

For each dependent variable, we fitted different response distributions and then made model comparisons to determine the best model using Leave-One-Out-Cross-Validation (similar to using Akaike Information Criteria in frequentist analysis). In each case, a skew-normal distribution proved to be the best-fitting model. In addition to calculating the probability of an effect above zero, defined as the proportion of the posterior distribution of the median’s sign, we also calculated the probability of a difference above a standardised difference of 0.254 was used. The reason this effect size was chosen is because it represents the value where 60% of the group with the higher estimated mean value were above the mean of the comparison group (Cohen’s U3), and is very slightly above Cohen’s “small” effect. For each regression model, estimated marginal means are reported in the results to allow the reader to make a straightforward comparison between the young BSA groups investigated.

## 3. Results

The descriptive characteristics of the main dependent variables in males and females across the four ethnic groups are presented in Table 1. These include: the sample sizes, age, reported seven-day activity levels (walking, moderate intensity, vigorous intensity and MVPA) and the percentage of each subgroup that meet the PA health guidelines. A wide range of differences in PA were evident between youth from the different ethnic groups and between males and females. 

As highlighted in Table 1, Pakistani participants spent the highest amount of time in vigorous PA and Bangladeshi participants the least. In contrast to differences found in vigorous PA, when comparing the self-reported moderate activity levels of the ethnic groups, Indian youth engaged in the most moderate PA and Afghan youth engaged in the least. When combining moderate and vigorous activity into total average MVPA, Indian youth spent the highest amount of time engaging in MVPA, while Bangladeshi participants spent the least amount of time participating in MVPA. These findings are illustrated in Figure 1. As highlighted in Table 2 and Table 3, there were differences across BSA groups, but the probability that most of these differences were above a small, standardised difference (d = 0.25) was generally very low. 

Estimated marginal means and differences between ethnic groups from Bayesian analysis are presented in Table 2 and Table 3. 

The values presented in Table 3 are the estimated marginal means of the minutes the different ethnic groups engaged in walking, moderate physical activity (MVPA), vigorous physical activity (VPA), and moderate-to-vigorous physical activity (MVPA). Estimated marginal means were extracted from the statistical model rather than the data and represent estimates of averages along with their uncertainty (the 95% Highest Density Intervals—HDI). Unlike traditional frequentist confidence intervals, points within the HDI have a higher probability density than points outside the interval—put simply, there is a 95% chance that the true population value is included in the interval.

Table 3 shows that, on average, there were minimal differences across the groups in terms of the activities detailed, except for MVPA for those from an Indian heritage.

A Bayesian Chi-square analysis of independence that modelled each cell count using a Poisson distribution suggested the data were 267 times more likely under a null hypothesis of no association in PA classifications of young people across genders or ethnic groups than an association. There was a very low percentage of each gender and ethnic group meeting the PA guidelines, as shown in Table 4. 

Table 4 shows pairwise contrasts for each of the different groups investigated in the study. The estimated differences presented are in minutes of MVPA engaged in. In column two, probabilities, expressed as a percentage, of differences between groups greater than a zero difference (labelled PD %) are presented. The final column presents the probability, expressed as a percentage, of difference between groups above a standardised difference > 0.254. Simply put, this is the chance that differences were greater than 0.254 standard deviations different between groups.

In contrast to the MVPA findings, Afghan youth reported spending the highest amount of time walking on average, with Indian participants spending the lowest amount of time walking (see Table 1). However, when accounting for the total energy expended (in METs) in all forms of activity (walking, moderate-intensity, and vigorous PA), the results were very similar to the above-reported findings of total time in MVPA. As outlined in Table 1 and illustrated in Figure 2, Pakistani males expended the highest weekly METs and Bangladeshi females expended the lowest active METs per week.

## 4. Discussion

The aim of this study was to assess the PA levels of young BSAs from different ethnic groups. This study also suggested that ethnic groups based in the UK originating from Afghan, Bangladeshi, Indian and Pakistani backgrounds are commonly treated as homogeneous because they share a common Asian heritage, but are heterogeneous in many respects, i.e., religious and cultural differences. The study highlighted an alarmingly low proportion of young people from each BSA ethnic group meeting the current guidelines of at least 60 min of MVPA per day [38]. Explicitly, only 2.3% of Afghans, 2.5% of Bangladeshi, 3.6% of Indians and 1.9% of Pakistani youth met the recommended minimum MVPA guidelines of 60 min per day. The latest national and regional statistics indicate that the percentage of young people in England who meet the PA guidelines is lowest in the West Midlands (42%) compared to all other regions in England. Moreover, the statistics also highlight clear ethnic disparities, with just 38.7% of Asian children and young people meeting the PA guidelines, compared to 47.7% of young people who identified as white British [9]. Alarmingly, the findings of the current study highlight even lower levels of BSA youth meeting the PA guidelines, underlining the urgent need to tackle inactivity in young people, and, in this instance in particular, in BSA youth in the West Midlands. Additional research is needed to further understand the barriers that these inactive groups of young people face to inform future targeted health promotion strategies.

The present findings support previous research focused on BSA young people, concluding that Bangladeshis were less active compared to Pakistanis and Indians [14,16,44]. The findings of this study also align with [8,45], suggesting that BSA young people engage in insufficient MVPA. Nonetheless, ref. [8] did not consider the BSA as heterogeneous and acknowledged them as homogenous, whereas the present study considered each ethnic group as heterogenic. In doing so, the current study highlighted some subtle differences in the PA levels of the different BSA groups, whilst fundamentally revealing very low PA levels across all groups (regardless of gender or ethnicity). To the authors’ knowledge, the current study is the first study to explore and compare PA levels of the British Afghan population, thus contributing novel findings to the area of BSA PA. 

The low levels of PA from Afghans, Bangladeshi and Pakistani young people may be partially attributable to religious (Islam) and cultural reasons. Concerningly, in the current study, no BSA females who identified as Afghan, Bangladeshi or Pakistani met the daily PA health guidelines, whereas the proportion of Indian females who met the guidelines was higher than males (7% and 0%, respectively). Islam is the dominant religion in all three of these ethnic groups (Afghans, Pakistani and Bangladeshi). Whilst Islam promotes active lifestyles regardless of gender, ability, or background, the lack of single-gender PA opportunities has typically been associated with lower PA levels within females from BSA communities that need population-specific solutions [44,46,47].

This extends beyond the Islamic religion, as the Sikh community, who are predominantly from Indian heritage, also exclude mixed-gender activities [48]. Furthermore, recent research highlights how some BSA families encourage young people to give priority to work and academic achievement over PA participation [47]. According to [49], parents from BSA ethnicities are encouraging their children to pay more attention to ‘academic subjects’ rather than Physical Education (PE) and after-school organised sports clubs. Thus, research shows that BSA young people have a negative approach towards PA and PE lessons, which may be due to lack of parental support and knowledge about the health benefits of PA [50]. Further literature indicates that parents can influence children to endorse healthy and active lifestyles [51]. The greater the parental PA levels, the more likely children are to meet the recommended PA guidelines [52]. Furthermore, there are also variations in PA levels among BSAs, as the second generation of BSAs appear to be more active than the first generation, but still less active than the White British community [7]. This suggests that providing education about the health benefits of participating in regular PA to BSA parents and family members may be pivotal in promoting PA within BSA communities.

Whilst the current study provides novel findings on the PA behaviours of the BSA community within an ethnically diverse region of the UK (the West Midlands), the findings are only reflective of young people within this specified region. Furthermore, whilst this study has highlighted disparities and similarities in the PA behaviours of youth from different BSA backgrounds, it does not explore the barriers that may contribute to these findings, nor does it consider the PA behaviours of BSA populations beyond the West Midlands. Therefore, future research is needed to further explore determinants of PA in a wider sample of BSA communities across the UK. Moreover, whilst it was beyond the scope of the current study to explore other lifestyle factors (such as sedentary or dietary behaviour and other variables that may also affect PA levels, such as parents being athletes, whether the family encourages them to perform PA, the free time they have available for PA, motivation, sleep, and BMI), further research in this area is warranted.

Whilst the IPAQ-SF represented a valid and practical data collection tool, it has been acknowledged in the previous literature that it is not without limitations [53]. There are inherent limitations in measuring PA with purely subjective methods, such as the underestimation or overestimation of time, often linked with social and recall bias [54]. Future research may wish to build upon the present study’s findings, with the use of objective PA measurement tools.

## 5. Conclusions

Overall, the vast majority of BSA young people failed to meet the recommended minimum MVPA guidelines of 60 min per day, with most achieving less than half the daily guidelines (80%). Whilst the proportion of youth meeting the PA guidelines was low across all ethnicities and genders, some subtle differences were evident between BSA groups according to PA intensity. Generally, average weekly MVPA levels were higher in Indian and Pakistani youth compared to Bangladesh and Afghan youth. In all ethnic groups except Indians, males reported being more active than females. These findings establish the similarities and differences in PA levels between different BSA groups, including the under-reported Afghan population in the UK. Moreover, the findings form a basis and rationale for further research, exploring the key determinants of PA behaviour within these inactive minority groups. Participating in insufficient levels of PA is a serious risk to health and could lead to cardiovascular diseases, obesity, stroke, and depression [50,55]. Thus, the present results are vital for UK communities, particularly for BSA ethnic minority groups, to consider engaging in PA and health-related interventions. Additionally, this study’s findings support the need for further PA research in BSAs according to individual ethnic group, particularly with the growing UK-based Afghan population who are currently an under-researched group. Finally, future research could utilise objective methods such as Global Positioning System, pedometers and accelerometery to track and monitor people’s PA levels, whilst also adopting an ecological approach to exploring determinants of PA within each ethnic and gender group. Such research will build on the existing findings and further contribute to tackling social health inequalities by informing future bespoke UK public health strategies.

## Figures and Tables

**Figure 1 ijerph-20-01087-f001:**
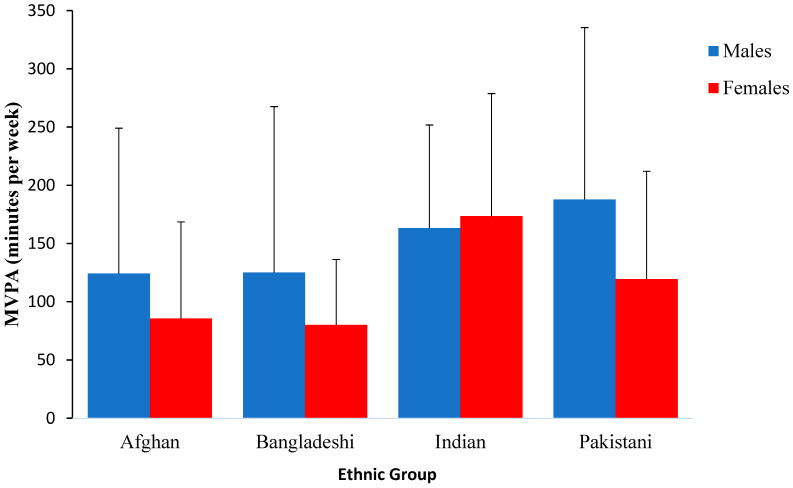
Total minutes of MVPA per week of males and females across the four ethnic groups.

**Figure 2 ijerph-20-01087-f002:**
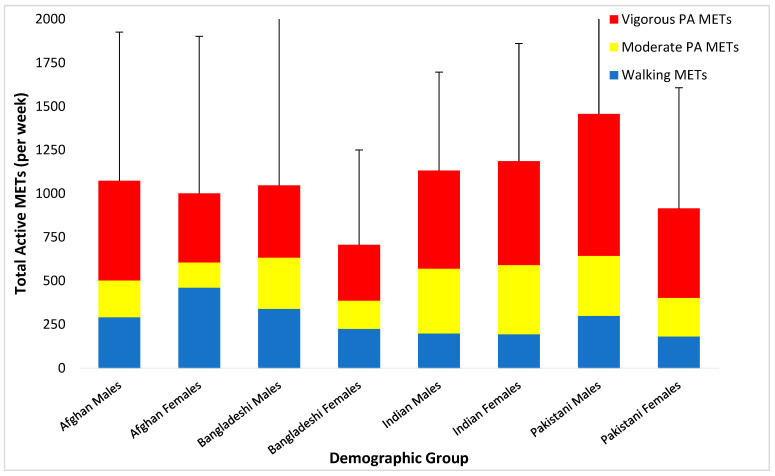
Total active METs per week of males and females across the four ethnic groups.

**Table 1 ijerph-20-01087-t001:** Means and standard deviations of the main dependent variables across the four ethnic groups.

	Afghans			Bangladeshi			Indians			Pakistani		
	Males	Females	Total	Males	Females	Total	Males	Females	Total	Males	Females	Total
N	26	18	44	19	20	39	28	28	56	25	27	52
Age (years)	15.3 ± 0.5	15.3 ± 0.5	15.3 ± 0.5	15.4 ± 0.5	15.5 ± 0.5	15.4 ± 0.5	15.5 ± 0.5	15.5 ± 0.5	15.5 ± 0.5	15.4 ± 0.5	15.5 ± 0.5	15.4 ± 0.5
Time spent walking (min/wk)	88.46 ± 99.0	139.72 ± 192.8	109.4 ± 145.0	102.75 ± 180.0	68.25 ± 131.9	85.5 ± 156.7	60.2 ± 34.6	58.9 ± 61.3	59.5 ± 49.3	90.74 ± 78.00	54.8 ± 75.9	73.5 ± 78.3
Moderate PA (min/wk)	124.5 ± 124.8	85.6 ± 83.0	45.9 ± 59.7	125.2 ± 142.4	80.2 ± 56.0	56.9 ± 73.7	163.2 ± 88.6	173.6 ± 105.0	96.0 ± 68.3	187.8 ± 147.6	119.5 ± 92.5	71.3 ± 81.0
Vigorous PA (min/wk)	71.4 ± 76.2	49.5 ± 70.0	62.4 ± 73.7	51.7 ± 69.9	40.0 ± 37.4	45.9 ± 55.6	70.8 ± 50.6	74.5 ± 43.8	72.3 ± 46.7	101.7 ± 86.6	64.0 ± 54.9	83.6 ± 74.8
MVPA (min/wk)	124.2 ± 124.8	85.6 ± 83.0	108.4 ± 110.2	125.3 ± 142.4	80.3 ± 56.1	102.8 ± 109.2	163.21 ± 88.6	173.6 ± 105.1	168.4 ± 96.5	187.8 ± 147.6	119.5 ± 92.5	154.9 ± 127.8
Total active METs	1074.1 ± 851.9	1001.5 ± 900.6	1044.4 ± 862.5	1047.1 ± 1327.9	706.2 ± 544.6	876.7 ± 1016.6	1457.2 ± 1005.2	915.1 ± 691.8	1196.6 ± 902.9	1132.16 ± 564.6	1186.6 ± 673.8	1159.4 ± 616.6
Meeting PA guidelines (%)	3.8%	0%	2.3%	5%	0%	2.5%	0%	7.1%	3.6%	3.7%	0%	1.9%

**Table 2 ijerph-20-01087-t002:** Estimated marginal means from Bayesian analysis of PA intensities across four ethnic groups.

	Afghans		Bangladeshi		Indians		Pakistani	
Activity	Est. Marginal Means	95% HDI	Est. Marginal Means	95% HDI	Est. Marginal Means	95% HDI	Est. Marginal Means	95% HDI
Walking	95	(80–108)	100	(87–113)	107	(93–123)	101	(88–115)
MPA	65	(53–76)	72	(62–83)	84	(73–97)	72	(63–82)
VPA	62	(52–71)	67	(58–76)	74	(65–85)	70	(61–79)
MVPA	118	(96–138)	131	(109–152)	159	(135–184)	138	(117–159)

**Table 3 ijerph-20-01087-t003:** Estimated differences of MVPA between ethnic groups.

Contrast	Estimated Difference	PD (%)	Probability Diff Above d = 0.254
Afghans—Bangladeshi	−11.4	100%	7%
Afghans—Indian	−40.2	100%	78%
Afghans—Pakistani	−18.17	100%	22%
Bangladeshi—Indian	−27.51	97.05%	46%
Bangladeshi—Pakistani	−5.67	68.75%	6%
Indian—Pakistani	21.28	94.47%	29%

**Table 4 ijerph-20-01087-t004:** Levels of PA of males and females across the four ethnic groups.

Ethnicity	Gender	N	Physical Activity Index		
			Low PA or Inactive (<30 Min/Day)	Insufficiently Active (30–59.9 Min/Day)	Active (>60 Min/Day)
Afghans	Males	26	88.5%	7.7%	3.8%
	Females	18	83.3%	16.7%	0.0%
	Total	44	86.4%	11.4%	2.3%
Bangladeshi	Males	19	80.0%	15.0%	5.0%
	Females	20	100.0%	0%	0%
	Total	39	90%	7.5%	2.5%
Indian	Males	28	78.6%	21.4%	0%
	Females	28	71.4%	21.4%	7.1%
	Total	56	75%	21.4%	3.6%
Pakistani	Males	25	63.0%	33.3%	3.7%
	Females	27	84.0%	16%	0%
	Total	52	73.1%	25.0%	1.9%

## Data Availability

Not applicable.

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
