# Peer review of "A Cross-Sectional Study Exploring the Physical Activity Levels of Afghans and Other South Asian Youth in the UK"

_ijerph, 2023, doi:10.3390/ijerph20021087_

Round 1
Reviewer 1 Report
Dear all,
Thank you very much for providing me the opportunity to review this manuscript. This research analysed the PA levels in South Asian population in UK. The study has as major strengths the solid scientific theoretical basis in the introduction and discussion, the statistics and the clearly presented results. Moreover, the inclusion of the Afghan population is a novelty. However, there are many papers analysing the PA levels in South Asians living in the UK and by gender. Before I can recommend the article for publication, few points need to be clarified.
Abstract
(I). Line 12. I recommend deleting "numerous" before chronic degenerative disorders to avoid the repetition of this word (it is used before: "numerous benefits").
(II) Line 13. Please, check if there is a missing comma after worldwide.
(III). Lines 14 and 15. Authors said: "The aims of this study were to explore 14 the PA levels of BSA" However, there is only one aim. I recommend writing the aim was... or include the aim of the analysis by gender.
(IV). The mean age of the participants with SD in the abstract is missing. Please, include it.
(V). Line 27. Please, check if its a double space prior "Future".
Introduction
(I). The references used are not current, please revise this section to include references from recent years.
(II). Please, check double spaces (lines 41 and 53).
(III). Line 37. Please, check if there is a missing comma after worldwide.
(IV). Line 69. Something is missing prior suggested, please correct it.
Methods
(I). Line 90. What was the Human Research Ethics Commission and the identifier number associated with this study?
(II). Line 93. Change "between...to" by "between...and"
(III). Please, check if there are double spaces (line 94, line 152 and line 160).
(IV). Was a validated instrument used to collect ethnographic, gender and other variables measured that did not include PA?
IV). Was the IPAQ-SF implemented in English? If so, how did you measure the participants' level of English language proficiency? On the other hand, I have not found validation of the IPAQ-SFQ in the Bangladeshi population.
(VI). Lines 108-109. Many studies claim that PA measured by IPAQ-SF does not correspond to PA measured by accelerometers. Authors should include that there is controversy on this issue and include some articles (examples: https://pubmed.ncbi.nlm.nih.gov/25252088/ https://pubmed.ncbi.nlm.nih.gov/28056080/)
(VII). Line 116. Please, check if there is a missing comma after activity.
(VIII). Line 120. Please consider change "pre and post prayer" by "pre- and post-prayer".
(IX). Line 121. The IPAQ-SF should only be implemented by the researcher, who knows the protocol and guarantees objectivity. Teachers and coaches may have a subjective point of view and interfere with the participants' answers. They are also not trained to implement the questionnaire.
(X). Line 124. Before writing acronyms, the whole word must be written. Thus it would be: Moderate to Vigorous Physical Activity (MVPA).
(XI). Line 133. The same: Metabolic equivalen of tasks (METs).
(XII). Lines 136. Delete the space in cut- off (it would be cut-off).
Results
(I). A descriptive table with demographic, ethnic, age and gender variables is missing before starting with the PA variables in order to know the characteristics of the participants. This should be included in the results and statistics section.
(II). Line 208 METs has been previously used in line 133.
Discussion
(I). As in the introduction section, the references should be revised to include more recent references.
(II). Please, check if there are double spaces (lines 223 and 256).
(III) Line 221, include a comma after respects.
(IV) Line 245. The authors highlight the inclusion of the Afghan population as a strength of the study but do not allude to its weaknesses. Weaknesses of the study should be included, some of them are:
- The small sample size.
- The data are not extrapolable to other regions of the UK, other countries, other ages and other South African populations.
- No objective PA measurement instrument such as accelerometers or pedometers has been used. Also include whether a validated instrument has not been used for ethnographic and gender measurements.
- Results may be contaminated by the subjectivity of teachers and coaches, who implemented the questionnaire.
- The absence of a measure of the English language level of the participants in case the questionnaire was in English.
- The IPAQ-SF is not validated for the Bangladeshi population.
- The lack of measurement of other variables that may affect PA levels (whether parents are athletes, whether the family encourages them to do PA, the free time they have available for PA, motivation, sleep, BMI, etc.), and the lack of measurement of other variables that may affect PA levels (whether the parents are athletes, whether the family encourages them to do PA, the free time they have available for PA, motivation, sleep, BMI, etc.).
Conclusion
(I) Please, check if there are double spaces (lines 282 and 285).
(II). Line 284, check if there is a missing comma after communities.
List of abbreviations
(I). Line 295. Delete WHO
References
(I) References are not adequate to IJERPH style in this section and along all the article. For example:
Journal Articles:
1. Author 1, A.B.; Author 2, C.D. Title of the article. Abbreviated Journal Name (italic) Year (bold), Volume, page range.
Please check all references be compatible with IJERPH style (ACS style).
Author Response
Dear Reviewer.
Thank you for reviewing our manuscript. I am pleased to have completed necessary amendments, and have uploaded this new version online as requested. I also attach/uploaded responses to your comments in the table below.
Please let me know if I can provide you with any further information. Thank you for your time.

Reviewer 2 Report
The topic is interesting and worth presenting. Issues are presented in understandable, logical way. However, there are some errors that should be corrected before publication:
First of all, does the study have the approval of the bioethics committee? If yes, please give the number and if not, the tests should not take place at all… In addition, the study material itself raises considerable reservations. “A total of 191 (Afghans N = 44; Bangladeshi N = 39; Indian N = 56, Pakistani 16 N = 52) participated in this study“ + “Afghans in particular 79 are one of the most rapidly increasing populations in the UK, raising from 63,000 in 2011 80 to an estimated 79,000 with 49% being British nationals”… Therefore, the examined group of 191 people, what it is the percentage of the entire BSA population? I mean, you wrote that Afghans are the most numerous 79,000, and you examined only 44 of them, which is less than 0.056% of the whole group. This is not a representative group. The number of observations is the basis for calculating the number of degrees of freedom for the analyzed tests, and thus has an impact on whether the obtained result will be statistically significant. The larger number, makes smaller difference between the examined groups turns out to be statistically significant. Therefore, it would be necessary to increase the research material or specify in the topic the group in terms of, for example, the region in which they live in the UK (West Midlands) or age (15-16 years). Think about it, because it's a big handicap of work.
79-84. “Afghans in particular 79 are one of the most rapidly increasing populations in the UK, raising from 63,000 in 2011 80 to an estimated 79,000 with 49% being British nationals (23).”
- Data from 2019 needs to be updated.
93. “BSA young people aged between 15 to 16 years of age provided consent to partake in the study.”
- They are minors, so the researcher should obtain written consent from their legal guardians. These are the ethical requirements of research.
137. “These cut- off points were applied to categorise young people’s PA in accordance with the current PA health guidelines”
- Tick UK guidelines or refer to WHO guidelines.
145-147. “Importantly for the present study, they do not require a randomised sample from the population of interest — something not attainable in cross-sectional studies.”
- This is debatable, because looking at the amount of BSA in England, the study group can be considered as randomized sample. Therefore, verify the figures contained in the introduction and determine what percentage of these tens of thousands are people aged 15-16. This will significantly increase the statistical significance of your group.
171-173. “These include: the sample sizes, 171 age, reported 7-day activity levels (walking, moderate intensity, vigorous intensity and 172 MVPA) and the percentage of each subgroup that meet the PA health guidelines.”
- A question arises in the reader's mind, did the respondents interpret the questions in the same way? That is, did they calculate time spent walking from planned or unplanned PA? Did everyone count only the walk that was extra, for health, or also the necessary one, such as getting to school/shop, running for the bus? If it is not important for the author, he should indicate in the introduction that he defines the general PA.
169. It would be worth adding a chart of the average activity of other UK residents to all the tables, so as to clearly see whether these differences exist and how large they are.
233-234. “Additional research is needed to further understand the barriers that these inactive groups of young people face, to inform future targeted health promotion strategies.”
- This is very important and should be emphasized in the text. Because from the beginning of reading the text there are thoughts about the reasons for this state of affairs. I would suggest that the author, as a subject specialist, dare to hypothesize why this is so and then suggest the need for additional research.
255-270. Very interesting insights worth developing. I miss the author's suggestion as a specialist why this is happening. Perhaps it is worth adding information about the economic situation of these parents and pointing out that in the minds of parents, education will bring money to their children's lives in the future, and PA will not, so they treat it only as some unnecessary entertainment. On the other hand, among affluent families there is often a lack of time with parents and they compensate for it with electronic toys for children, not teaching them from an early age the need for PA. Such data exists, it is worth the author to mark it in this article.
271. In the conclusions, it is worth mentioning what is the position of the British Government on this matter and whether they are taking any actions to activate new UK residents
Despite the above comments, which are aimed at improving the quality of work, I think the article is interesting, important and worth submitting after minor revision.
* As a person whose first language isn't English, I don't feel competent to evaluate the grammar and style of a text.
Author Response

(The authors gave the same response as above.)

Reviewer 3 Report
The manuscript entitled " A cross-sectional study exploring the physical activity levels of
Afghans and other South Asian youth in the UK" are an important study that could fill the gap in the literature on this topic. Generally, the study is well written, however, the methods and the results part need some revision and restructuring to help the readers to understand the study.
My suggestions to the Authors:
· The introduction is well-written. The reader can understand the physical activity level for ethnic groups. However, the study is about young BSA people, therefore it would be beneficial to write about this age group's physical activity as well.
· Please provide more information on the procedure and how data collection accrued.
· How minutes/week was calculated
· I believe lines between 122 and 137 seem like a limitation, which should be addressed in the limitation section.
· In the result section, table 4: How do you determine low PA? Please add this information to the methods.
· I believe the results section is generally hard to follow. I suggest writing a little more on the table and figures since in some descriptions the readers can't really understand what figures the authors describe. (e.g., Line 192-193).
· Table 1-3 is next to each other. Please separate them.
· Add limitations to the discussion
Author Response

(The authors gave the same response as above.)

Round 2
Reviewer 1 Report
Thank you to the authors for addressing my concerns and for the detailed response. I think the paper is now well-written, and has successfully overcome the difficulties inherent to the study. However, I have some minor points:
Methods
- I hope that the researchers' help to the participants with any misunderstandings of the IPAQ did not interfere with the results and data obtained.
- Lines 115-116. Authors should delete "studies comparing the validity of the IPAQ to accelerometers have demonstrated an acceptable level of reliability in measuring PA patterns" or inculde that some controversy exists on this regard.
Results
- Line 254. Please, delete "." after 0.254..
Discussion
Authors should include the following items in the limitations section:
- The non-use of a validated instrument to measure ethnographic variables. This tool should be taken into account in future research.
- The lack of measurement of other variables that may affect PA levels (whether parents are athletes, whether the family encourages them to do PA, the free time they have available for PA, motivation, sleep, BMI, etc.), and the lack of measurement of other variables that may affect PA levels (whether the parents are athletes, whether the family encourages them to do PA, the free time they have available for PA, motivation, sleep, BMI, etc.).
References
References are not in IJERPH style (ACS style).
Author Response
Dear Reviewer.
Thank you for reviewing our manuscript. I am pleased to have completed necessary amendments, and have uploaded this new version online as requested. I also attach/uploaded responses to your comments in the table below.

Reviewer 3 Report
Thank you for the Author's contribution to this study. All changes have been made. I recommend publishing.
Author Response
Dear Reviewer,
Thank you.
